# Design and Analysis of a Chinese Medicine Based Humanoid Robotic Arm Massage System

**Zaixiang Pang [1], Bangcheng Zhang [1,\*], Junzhi Yu [2,\*] , Zhongbo Sun [3] and Linan Gong [4]**

[1] School of Mechatronical Engineering, Changchun University of Technology, Changchun 130012, China; pangzaixiang@ccut.edu.cn

[2] State Key Laboratory for Turbulence and Complex System, Department of Mechanics and Engineering Science, BIC-ESAT, College of Engineering, Peking University, Beijing 100871, China

[3] College of Electrical and Electronic Engineering, Changchun University of Technology, Changchun 130012, China; zbsun@ccut.edu.cn

[4] Changchun Vocational Institute of Technology, Changchun 130000, China; gonglinan2019@163.com

\* Correspondence: zhangbangcheng@ccut.edu.cn (B.Z.); yujunzhi@pku.edu.cn (J.Y.);
Tel.: +86-133-3175-5427 (B.Z.); +86-136-9105-5851 (J.Y.)

**Abstract:** This paper presents a humanoid robotic arm massage system with an aim toward satisfying the clinical requirements of pain relief on the waist and legs of older patients during Chinese medicinal massage. On the basis of an in-depth analysis regarding the characteristics of arm joints of the human body and Chinese medicinal massage theory, a humanoid robotic arm massage system was designed by adapting a bottom to top modular method. The combined finite element and kinematic analysis led to an improved performance according to repeated positioning accuracy, massage strength accuracy, and massage effect. The developed humanoid robotic arm was characterized by a compact structure, high precision, light quality, and good stiffness, achieving a good bearing capacity. Due to the PID controller, the numerical simulations and experimental results provided valuable insight into the development of Chinese medicinal massage robots and massage treatments for patients who suffer from lumbar muscle strain.

**Keywords:** humanoid; Chinese traditional massage; robot arm; design and analysis

## 1. Introduction

According to the revised report of the 2017 world population prospects released by the United Nations, current trends in the world's aging population will continue to grow over the next 30 years. The world's elderly population is expected to triple to 2.1 billion by 2050 and will result in a considerable demand for medical and health care services, which has become a major social problem [1]. Undoubtedly, service robots can relieve the shortage of medical personnel, improve the quality of life for the elderly and the disabled, and play a positive role in the stable development of China [2]. Currently, various degenerative diseases and the chronic pain which often accompanies them have become major ailments in older populations, among which lumbocrural pain is the most common and severe.

Owing to the long history of Chinese massage therapy, traditional Chinese medicinal massages are effective and efficient tools to relieve and treat most kinds of chronic pain. According to natural healing theory, Chinese medicinal massages are a type of health care massage for chronic pain that was developed for more than 2000 years ago. It advocates for natural massage as a means to strengthen the immune system of the human body and guide the natural rehabilitation of patients. However, major clinical treatments at hospitals are still being performed through the use of manual labor,

generally leading to a considerable waste of resources. Nowadays, various massage practitioners are currently marketed for patients and hospitals. However, a massage therapist's singular approach can only provide relaxation and alleviate fatigue for patients, which hardly achieves the aim of comprehensive treatment.

A summary of massage processes and techniques in clinical applications are presented in this paper. In the process of promoting massage, the key influences of massage cannot be accurately and quantitatively described using human language, including massage position, strength, manipulation, and a patient's psychological adjustment, which will therefore affect the promotion of massage techniques in hospital settings. Moreover, a humanoid robotic arm massage system can imitate the actions of a masseuse which causes no fatigue and can effectively alleviate the lack of massage staff.

Many companies and institutions have conducted relevant research on robotic massage systems. Massage service equipment has been developed which has expanded the scope of care and treatment of massage [3]. Force control for path planning of induction-based massage robots was investigated and analyzed [4]. Moreover, a three degrees of freedom (DOFs) massaging robot was designed and investigated for patients [5]. Chinese traditional massage mainly emphasizes the stimulation of manipulation and changing power on a series of meridians which improves the ability of the immune system and achieves the purpose of disease prevention and treatment. There are several works which have extensively studied the finger and joints of the human hand for use in a multi-finger massage robot [6–8]. Existing massage devices and robots are generally designed for only certain massage techniques and can only realize one or two different massage patterns for patients. Subsequently, a novel PUMA562 robot platform was developed and designed for people [9]. Recently, some classical massage robots have been developed, such as the WAO-1 [10,11], a humanoid massage robot [12], a four-finger human hand robot [13], and a robotic hand [14]. To relieve lumbago and leg pain for middle/old-aged patients suffering from degenerative diseases, a Chinese-style massage robot was designed and investigated on the basis of Chinese medicinal massage theory [15–17]. The massage force can be actively controlled through force sensors; however, a personalized massage cannot be implemented for patients. In addition, Hu et al. [18] developed and analyzed a Chinese massage robot which featured an arm with four DOFs. This robot had the end of its arm in a series with three degrees of freedom, and a parallel wrist with three degrees of freedom served as the human hand which massaged. Additionally, two fingers were ineffective and the other fingers could undertake actions such as clicking, finger kneading, pinching, and rolling. Therefore, the five traditional kinds of actions which comprise Chinese medicinal massage techniques could be performed [19]. Furthermore, an increasing number of researchers are studying humanoid robot arms from the perspective of service robots [20–25]. Therefore, Chinese traditional massage devices and massage robots can be adjusted passively by the masseur or the massage object, which makes it difficult to ensure the accuracy and reliability of the massage position.

In recent years, much attention has been paid to massage equipment and massage robotic systems, which can be applied to many fields such as health care and rehabilitation therapy. The technical and functional characteristics of massage robot arms are as follows: (1) a relatively high degree of humanity and (2) an ability for remarkably complex movements and accurate grasp of massage force. In particular, a humanoid massage robot arm is rarely focused on configuration for realizing various massage techniques and precise recognition of a designated massage position. Furthermore, during the performance estimation of a humanoid Chinese medicinal massage robotic arm system, many factors including the weight of the arm and massage force size, direction, location, acupuncture point, and frequency should be taken into account. Therefore, a robotic arm with rigidity, high transmission accuracy, and anthropomorphic degree is highly desirable for the patients.

To address the abovementioned shortage of medical personnel and improve the quality of life of the elderly and the disabled, a humanoid robot massage arm was designed on the basis of human arm massage theory. In this paper, a humanoid massage robot arm was mainly used in the field of traditional Chinese medicinal massage to realize massage of the lumbar spine and back. According

to the task function and control requirements, the humanoid massage robot arms were designed with the joint as the module based on the characteristics of a human skeleton and using massage theory according to traditional Chinese medicine, contrary to previous humanoid massage robot arms. The joints of the humanoid massage robot arm were connected together to realize the basic functions of a human-like arm by taking advantage of the working space and good flexibility of the serial robot. Binocular vision positioning technology was utilized to determine the precise massage position. Furthermore, a three-finger, dexterous massage hand and parallel robot massage were used to reproduce the techniques of experts. Moreover, the movement of the robot arm drove the dexterous hand and the parallel robot massage terminal to the designated position. In addition, the humanoid massage robot massaged the patient through the massage terminal at the front of the arm and the massage hand. Therefore, it was able to achieve the massage techniques (which can be considered as press, knead, pinch, roll, vibration) on the human lumbar spine and the back of the palm which stimulates meridian points and improves the ability of the immune system. Therefore, the natural posture of the arm of the humanoid Chinese medicinal massage robot is an important indicator of a massage robotic system [26]. Four degrees of freedom in a humanoid Chinese medicinal massage robotic arm (through serial structure in a series with the elbow, forearm, and wrist joints) can easily achieve this action with the arm.

The mechanical structure of the humanoid massage robot arm was developed, investigated, and analyzed through a modular design concept, which was composed of a shoulder joint, elbow rotatory joint, wrist swing, and wrist rotatory joint. First, the motor of shoulder joint was installed perpendicular to the axis of the arm, which increased the motion space of the arm and provided convenience for replacing the motor. Second, a U-like structure was adopted in the shoulder bracket to improve the bearing capacity of the arm. Third, the motor (which was installed in the arm) was coincident with the center line of the arm to improve the transmission precision and reduce the diameter of each joint. Therefore, the characteristics of the humanoid massage robot arm were a compact structure and anthropomorphism, as well as an assembly which reduced the accumulated error of the arm. Bevel gear was utilized in the swinging joint, which improved the stability of the humanoid massage robot arm. A servo motor was used to drive each joint of the humanoid massage robot arm, and a reducer and brake were installed to ensure overload protection and emergency braking.

The strength and stiffness of the humanoid massage robot arm were analyzed to verify that the mechanical mechanism satisfied the design requirements of the mechanics. The kinematic simulation of the humanoid massage robot arm was illustrated and carried out through the establishment of a virtual prototype model, which provided a reference for the improvement of the humanoid massage robot arm.

A three-loop servo system was established, investigated, and verified for the humanoid massage robot arm. The smooth operation and trajectory positioning of the humanoid massage robot arm were realized by adjusting the PID (Proportion-Integral-Derivative) controller of the humanoid massage robot arm to adjust the parameters in combination with the PMAC (Programmable Multi-axis Controller) motion controller.

## 2. Mechanism Analysis of the Humanoid Massage Robot Arm

### 2.1. Motion Analysis of Commonly Used Chinese Massage Techniques

The arm of a humanoid massage robot is mainly aimed at treating chronic and degenerative diseases in middle-aged and elderly people. Generally speaking, doctors usually perform pressing, kneading, and rolling, which are traditional Chinese massage techniques [18]. However, the operation of a robotic arm on the platform of a massage robot facilitates massage on the lumbar vertebra and back of the human body by pressing, pinching, kneading, rolling, vibration, and other kinds of massage. This context focused on a relaxed lumbar massage for the patients. Therefore, relaxation from lumbar

massage was utilized from traditional techniques for motion analysis. Moreover, the movement state analysis was analyzed and investigated through common Chinese massage techniques.

To analyze the kinematics and force features of the massage robot, as shown in Figure 1, a coordinate system was established and investigated. Owing to the right-hand rule, the *z*-axis was perpendicular to the treatment surface, the *x*-axis parallel to the body surface direction, and the massage direction of the head positive to the *y*-axis. Moreover, the massage techniques used for the patients were categorized into the rolling technique, the thumb kneading technique, the pinching technique, the pressing technique, and the vibrating technique according to the kinematic space.

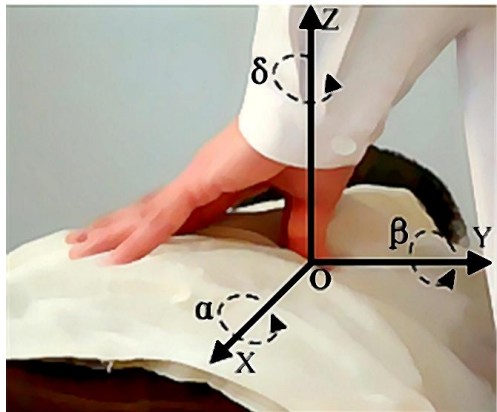

**Figure 1.** A massage technique's coordinates at the working spot.

### 2.2. Design and Mechanism Analysis of the Humanoid Massage Robot Arm

The joint characteristics of the human arm show that th ehuman arm contains a total of 7 DOFs, including the shoulder joint (3 DOFs), elbow joint (2 DOFs), and wrist joint (2 DOFs), as shown in Figure 2.

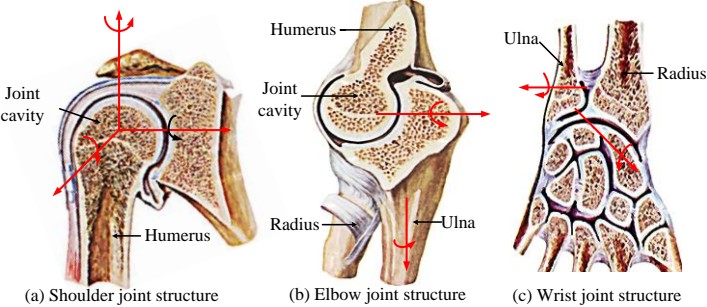

(a) Shoulder joint structure　　　(b) Elbow joint structure　　　(c) Wrist joint structure

**Figure 2.** Schematic diagram of arm joint movement.

The robot arm designed in this paper was mainly applied to human body massage. The massage robot system was mainly composed of a rehabilitation massage bed platform, gantry mobile platform, robot arm, and end actuator. The robot arm was mounted on the mobile platform. Through the analysis of movement during Chinese medicinal massage, it was determined that when the massage technique is performed, only four degrees of freedom are required to complete the manipulation of pressing, pinching, and plucking. Moreover, in the three normal kinds of manipulation operations described above, the wrist of the physician does not move, and the actuator is mainly utilized at the end of the robot arm to perform the massage operation. The main function was to cooperate with the position information provided by the binocular vision positioning module in the robot system and deliver the end effector to the correct human body massage position, which supported the end effector and, finally, the corresponding end effector.

The robot arm adopted a series structure, used an articulated structure, and was divided into the structure of a big arm, an arm, a wrist, etc. The arm completed one rotation degree of freedom and realized the lifting of the arm, while the arm completed one rotation degree of freedom, which drove the wrist. The joint and the end actuator rotated and the wrist both completed one rotation degree of freedom, thereby realizing the rotation and lifting of the end effector, respectively. Since the robot arm could be mounted on a moving platform with additional degrees of freedom, the end effector of the arm could achieve any position in the space.

The strength of the massage arm is more than 5 kg. The accuracy of joint position is less than 2 mm, and the error of massage force is less than 10%. Figure 3 shows the mechanical schematic diagram of the positioning platform. Further, Figure 4 illustrates a humanoid massage robot arm system consisting of a massage robot, a massage bed, a computer operating system, a visual tracking system, and auxiliary devices. The detailed body parameters are listed in Table 1. An adjustable and lightweight arm is fully considered during the design process. The process of rehabilitation training should also be considered for the safety of massage operation. Therefore, the arm joints in the installation of torque limiting agencies will provide automatic overload protection, and the arm outside the installed limit switch will improve the security of the system.

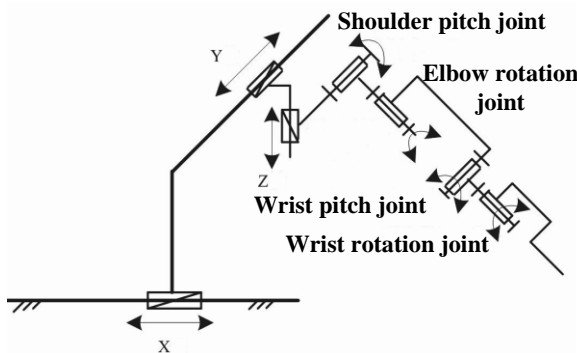

**Figure 3.** Mechanical schematic diagram of the positioning platform.

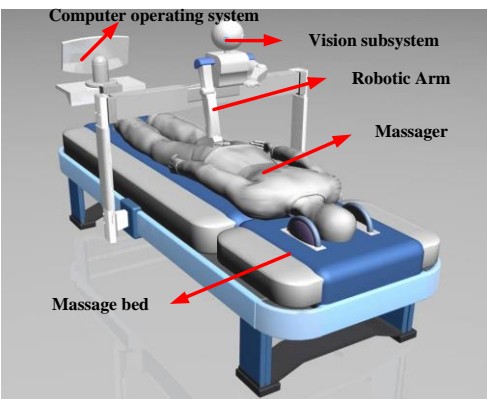

**Figure 4.** System configuration of the massage robot.

**Table 1.** Body parameters of the humanoid massage robotic arm.

| Degrees of Freedom | Arm Weight/kg | Maximum Outside Dimension/mm | Allowed Maximum Static Load/kg | Repositioning Precision/mm | Maximum Working Space/mm | Maximum Motion Range |
|---|---|---|---|---|---|---|
| 4 | 16 | 500 × 140 × 140 | 10 | ±0.2 | 1000 × 400 × 300 | Big arm pitch, ±90° Elbow rotation, 0–360° Wrist pitch, ±90° Wrist rotation, 0–360° |

The principle, technological implementation, and application are three aspects that should be considered to ensure the safety of the operation of the massage robot platform. The human–machine interaction interface allowed the masseur to monitor the massage process and adjust the massage program at any time by using the touching screen. In the principle layer, the applicability and safety of the massage operation provided to the patient could be completely ensured through the selection of appropriate massage techniques and processes. In the technical implementation layer, overload protection was realized with a device limiting mechanical force. A physiological signal monitor could determine an abnormal physiological state of the patient's arm and automatically stop the massage operation. The voice control and manual control provided the patients and the masseur with manual intervention in the massage process. Thereby, it can guarantee the safety of the system for the patients. In the application layer, the establishment of a sound system using a process and personnel training mechanism facilitated the analysis and evaluation of the possible risk. A preventive method would ensure the safety of the system during the process of rehabilitation training.

## 3. Kinematic Analysis of the Humanoid Massage Robot Arm

### 3.1. Kinematic Positive Solutions

In this subsection, the kinematic relationship between the joints of the robot and rigid bodies of the humanoid massage robot arm is described. The motion of forces and moments are not taken into account in this paper which involves the position, velocity, and acceleration of the moving object [27]. Kinematic analysis is an important basis for motion control, trajectory planning, and position control of the robotic arm. A forward kinematic mainly addresses the problem of the establishment of a motion equation and determines hand position. Figure 5 shows a schematic diagram of the kinematic coordinate system which is established using the $D-H$ method of the humanoid massage robot arm. Here, $L_1$ denotes the length of the upper arm and $L_2$ is the length of the cubitus. There are 4 DOFs for the humanoid massage robot arm, i.e., $\theta_1$ is the rotation angle of the upper arm, $\theta_2$ is the rotation angle of the elbow, $\theta_3$ is the rotation angle of the wrist, and $\theta_4$ is the pitch angle of the wrist. The origin $O_3$ coincides with the origin $O_4$, and $O_5$ is a hand center coordinate. The $D-H$ parameters and joint variables are presented in Table 2 for the humanoid massage robotic arm. The transformation matrix of the $D-H$ method for the adjacent coordinate system is generalized as follows.

$$A_i = \begin{bmatrix} C_i & -S_i C\alpha_i & S_i S\alpha_i & a_i C_i \\ S_i & C_i C\alpha_i & -CiS\alpha_i & a_i S_i \\ 0 & S\alpha_i & C\alpha_i & d_i \\ 0 & 0 & 0 & 1 \end{bmatrix} \tag{1}$$

where $C_i = \cos\theta_i$, $S_i = \sin\theta_i$, and $C\alpha_i = \cos\alpha_i$, $S\alpha_i = \sin\alpha_i$.

**Table 2.** $D-H$ parameters and joint variables of the humanoid massage robot arm.

| Link | Variable | $\alpha_i$ | $a_i$ | $d_i$ | $\cos\alpha_i$ | $\sin\alpha_i$ |
|------|----------|-----------|-------|-------|----------------|----------------|
| 1 | $\theta_1$ | 90° | 0 | $d_1$ | 0 | −1 |
| 2 | $\theta_2$ | 90° | 0 | $d_2$ | 0 | 1 |
| 3 | $\theta_3$ | −90° | 0 | $d_3$ | 0 | −1 |
| 4 | $\theta_4$ | 0 | 0 | 0 | 1 | 0 |

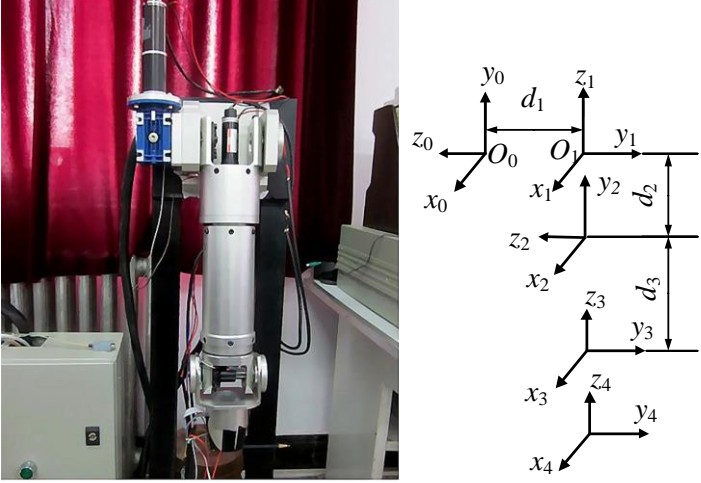

**Figure 5.** Kinematics diagram of the robot arm using the *D–H* method.

If the parameters in Table 2 are substituted into Equation (1), the transformation matrix $A_i$ can be expressed as follows:

$$A_1 = \begin{bmatrix} C_1 & 0 & -S_1 & 0 \\ S_1 & 0 & C_1 & 0 \\ 0 & -1 & 0 & d_1 \\ 0 & 0 & 0 & 1 \end{bmatrix}, A_2 = \begin{bmatrix} C_2 & 0 & S_2 & 0 \\ S_2 & 0 & -C_2 & 0 \\ 0 & 1 & 0 & d_2 \\ 0 & 0 & 0 & 1 \end{bmatrix}, A_3 = \begin{bmatrix} C_3 & 0 & -S_3 & 0 \\ S_3 & 0 & C_3 & 0 \\ 0 & -1 & 0 & d_3 \\ 0 & 0 & 0 & 1 \end{bmatrix}$$

and

$$A_4 = \begin{bmatrix} C_4 & -S_4 & 0 & 0 \\ S_4 & C_4 & 0 & 0 \\ 0 & 0 & 1 & 0 \\ 0 & 0 & 0 & 1 \end{bmatrix}$$

Therefore, the end position matrix is described as follows:

$$T_4 = A_1 A_2 A_3 A_4 = \begin{bmatrix} C_4 & -S_4 & 0 & 0 \\ S_4 & C_4 & 0 & 0 \\ 0 & 0 & 1 & 0 \\ 0 & 0 & 0 & 1 \end{bmatrix} = \begin{bmatrix} n_x & o_x & a_x & p_x \\ n_y & o_y & a_y & p_y \\ n_z & o_z & a_z & p_z \\ 0 & 0 & 0 & 1 \end{bmatrix} \quad (2)$$

where

$n_x = c_4(s_1 s_3 c_1 c_2 c_3) c_1 s_2 s_4;$
$n_y = c_4(c_1 s_3 + s_1 c_2 c_3) - s_1 s_2 s_4;$
$n_z = -c_2 s_4 - s_2 c_3 c_4;$
$o_x = s_4(s_1 s_3 - c_1 c_2 c_3) - c_1 s_2 c_4;$
$o_y = -s_4(c_1 s_3 + s_1 c_2 c_3) - s_1 s_2 c_4;$
$o_z = s_2 c_3 s_4 - c_2 c_4;$
$a_x = -s_1 c_3 - c_1 c_2 s_3;$
$a_y = c_1 c_3 - s_1 c_2 s_3;$
$a_z = s_2 s_3;$
$p_x = d_3 c_1 s_2 - d_2 s_1;$
$p_y = d_2 c_1 + d_3 s_1 s_2;$
$p_z = d_1 + d_3 c_2.$

where, *n*, *o*, and *a* represent the postures of the base coordinate, and *p* represents the position of the base coordinate.

### 3.2. Inverse Kinematic Solution

According to the position of the hand, the angle of each joint should be determined in practice. That is, we should determine the inverse kinematic solution of the arm of the humanoid Chinese medicinal massage robot. The inverse kinematic solution of the arm of the humanoid Chinese medicinal massage robot was the mechanism to determine the joint variable in the case of a known target position. Therefore, the inverse kinematic solution was the basis of arm movement planning and trajectory control for the humanoid Chinese medicinal massage robot. The joints of the motor were manipulated to obtain the position of the hand. The dexterous characteristics of the massage robotic arm were general multi-solutions for the process of rehabilitation training. Indeed, the result of the inverse trigonometric function equation causes the various solutions to the inverse kinematics of the massage robot arm. The massage robotic arm may be solved for only one set of solutions in an actual working environment. Furthermore, an appropriate assessment and solution are necessary in actual situations. The inverse kinematics of a 4 DOFs humanoid robot arm can be solved through the following equations.

The term $A_1^{-1}$ should be multiplied by the terminal position matrix $T_4$, then:

$$A_1{}^1 T_4 = A_2 A_3 A_4 \tag{3}$$

According to the principle of equal matrices, the following equations can be obtained as:

$$p_y c_1 - p_x s_1 = d_2 \tag{4}$$

$$p_z - d_1 = d_3 c_2 \tag{5}$$

$$s_2 s_3 = a_z \tag{6}$$

$$o_x s_1 - o_y c_1 = s_3 s_4 \tag{7}$$

$$\text{Let } p_x = \rho \cos \phi; p_y = \rho \cos \phi \tag{8}$$

where $\rho^2 = px^2 + py^2$; $\phi = A \tan^2(p_y, p_x)$; $\rho^2 = px^2 + py^2$.

Substituting Equation (7) into Equation (3) yields the following equations:

$$\sin(\phi - \theta_1) = \frac{d_2}{\rho}; \cos(\phi - \theta_1) = \pm \sqrt{1 - \frac{d_2{}^2}{\rho^2}};$$

and

$$\phi - \theta_1 = A \tan^2(\frac{d2}{\rho}, \pm \sqrt{1 - \frac{d_2{}^2}{\rho^2}}).$$

Then $\theta_1$ can be calculated as follows:

$$\theta_1 = A \tan^2(p_y, p_x) - A \tan^2(d_2, \pm \sqrt{px^2 + py^2 - d_2{}^2}) \tag{9}$$

and $\theta_2$ can be calculated by using Equation (4). The details can be seen as:

$$\theta_2 = arc \cos[\frac{(p_z - d_1)}{d_3}] \tag{10}$$

$\theta_3$ can subsequently be obtained by substituting Equation (9) into Equation (5).

$$\theta_3 = arc \sin(\frac{a_z}{\sin \theta_2}) \tag{11}$$

Finally, $\theta_4$ we can obtain via substituting Equations (8) and (10) into Equation (6).

$$\theta_4 = arc\sin(\frac{o_y \cos \theta_2 - o_x \sin \theta_1}{\sin \theta_3}) \tag{12}$$

### 3.3. Workspace Analysis

To determine the working space at the end of the massage mechanical arm with the angle of each joint as input function, the corresponding system model was established based on the MATLAB SIM Mechanics toolbox, and conducted through the motion simulation. The rotating joint only adjusted the posture of the end-effector and did not affect the working space at the end of the humanoid Chinese medicinal massage robot [28]. Hence, the rotating joint can be regarded as a simplified version when the mechanical system model was established for the humanoid Chinese medicinal massage robot arm. The workspace at the end of the arm of the massaging robot was determined by considering the rotation angle of each joint as the input, as shown in Figures 6 and 7. The workspace of the humanoid Chinese medicinal massage robotic arm was similar to the 3/4 cylinder with the shoulder joint as the axis. Apparently, this workspace could satisfy the requirements of a massage operation.

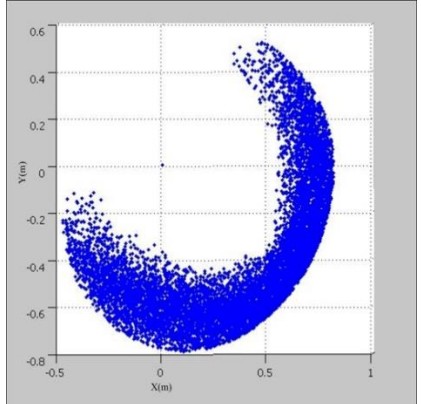

**Figure 6.** Workspace at the end of the arm (2D).

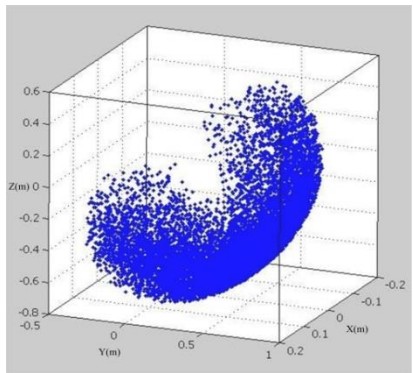

**Figure 7.** Workspace at the end of the arm (3D).

## 4. Modular Design of the Humanoid Robotic Arm Massage System

To reasonably allocate dimensions and ensure the coordination of dimensions and smooth progress, it was necessary to check whether the parts touched or blocked each other, which may result in the failure of normal installation. In addition, it was necessary to determine whether the positioning and assembly relationship among the parts was reasonable and whether the overall structure was aesthetically pleasing for the humanoid massage robot arm. The 3D solid modelling of the humanoid massage robot arm was carried out and investigated based on CATIA 3D modelling software, which

realized parametric modelling and virtual assembly of the humanoid massage robot arm from top to bottom. It established an accurate analysis model for finite element analysis, which improved the speed and accuracy of the design requirements.

On the basis of the characteristics of the human arm and Chinese massage theory, we considered the driving torque, moving speed, weight and size of the arm, direction, position, acupuncture point, and frequency of the human body. According to the design methods, we selected the appropriate combination of modules into the arm, namely, the shoulder, elbow rotation, wrist pitch, and wrist rotation joints. The modular articulation unit of the arm consisted of a motor, a reducer, a feedback device, and other accessories. The joints exhibited relative independence, and all joints worked together to achieve the basic functions of the humanoid massage robotic arm. The arm of the joint module shell mainly supported the connection motor, reducer, bearing, and other internal parts, which played a protective role. The shell adopted was an aviation aluminum alloy with a cylindrical shape, sufficient strength and rigidity, an anodized outer surface, and good appearance. The modules were relatively independent of each other. The various modules could be connected together to achieve the function of a humanoid massage robotic arm. Therefore, the connection of the configuration of the robot arm played an important role in imitation. The quality of the connector also affected the weight of the entire arm of the massage robot. Thus, mechanical strength should also be considered for humanoid robotic arm massage systems. Owing to the low density, high strength, and good plasticity, aviation aluminum alloy was utilized as the connection material. The resulting modular joints are shown in Figure 8.

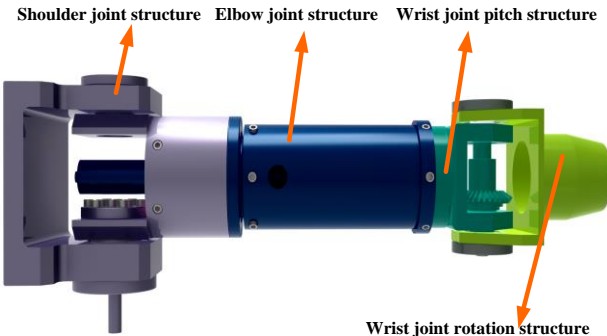

**Figure 8.** 3D model of the shoulder, elbow, and wrist joints.

The end-effector of the humanoid massage robot arm mainly consisted of a three-finger massage dexterous hand for the manipulation of kneading and a miniaturized parallel robot massage terminal for the manipulation of pressing, kneading, vibration, and rolling. The massage marker points needed to wear vests and were prone on the massage bed through the binocular visual positioning sensors before implementation of massage. The movement of the humanoid massage robot arm drove the dexterous hand massage and parallel robot terminal to the location and, combined with the arm front terminal, massaged the hand of the human body, as shown in Figure 9.

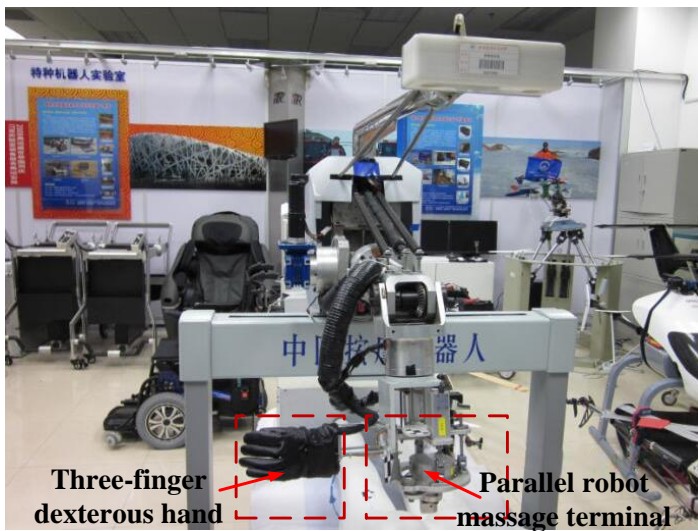

**Figure 9.** The end-effector of the humanoid massage robot arm.

## 5. Massage Robot Arm Control System Design

According to the designed requirements of the massage robot arm, the hardware system mainly included the mechanical structure, a PC, a motion controller programmable multi-axis controller (PMAC), an input/output (I/O) card, a Copley digital servo driver, a brushless DC motor. The massage robot system adopted the second-level control mode, and the PC communicated with the PMAC motion control card by transmitting 100 Mbps via Ethernet. The overall block diagram of the control system is shown in Figure 10.

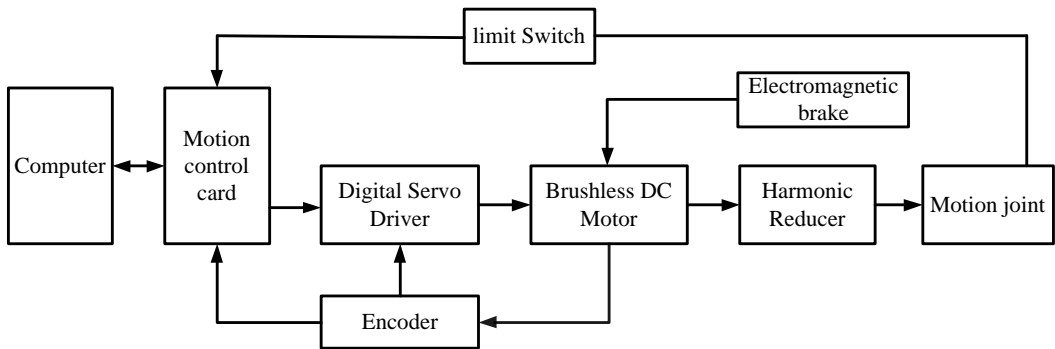

**Figure 10.** A block diagram of the massage robot control system.

### 5.1. Design of a Three Closed-Loop Servo System for a Massage Robot Arm

In order to improve the massage efficiency and quality, the massage robot arm had a higher requirement on positioning of the massage acupuncture points of the massaged person and the force control precision exerted on the acupuncture points by the end-effector. The system used a DC servo three closed-loop system to improve the control quality of the massage robot arm, which included position loop, speed loop, and current loop.

The three closed-loop system is shown in Figure 11. The current loop was the inner loop of the entire control system and was implemented inside the driver. The function of the current loop was mainly to adjust the motor current to exhibit the desired torque characteristics by the motor output torque. The current loop was also called the torque loop, which was given as the output of the speed regulator, and the feedback current sampling was done inside the driver. The speed loop was also the inner loop of the entire control system, which was outside the current loop and within the position loop.

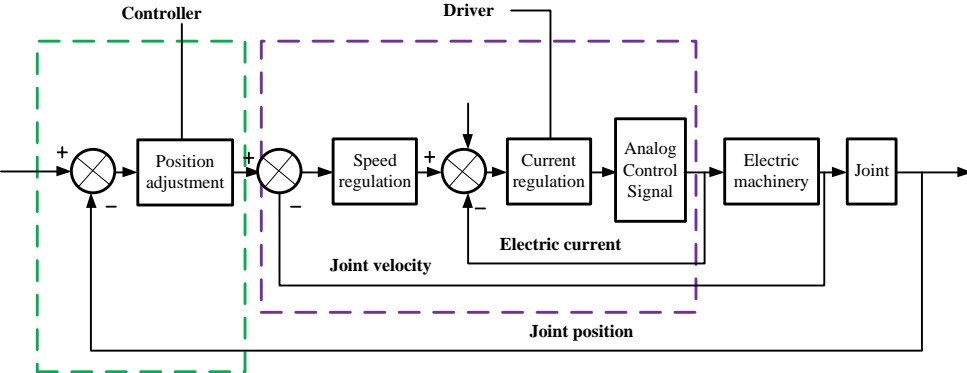

**Figure 11.** Single joint motor servo control schematic.

The loop was achieved on the PMAC motion controller, and the speed and current loop were achieved on the Copley digital servo drive. The Ethernet network was utilized to communicate with the PMAC motion controller, and the PEWIN32 software was adopted to set and debug the PID parameters of the joint motor.

## 5.2. Design PID Controller for the Servo Control System

The computer adjusted the corresponding I variable through the executable program PEWIN32 to realize the parameter, speed, and acceleration feed forward parameters of the PID controller. The parameters of the notch filter were adjusted, and the speed feed forward signal and the acceleration feed forward signal were designed based on the control algorithm of the PMAC motion controller. The effect of the speed feed forward was to reduce the following error caused by the differential gain in the PID controller. The effect of the acceleration feed forward compensated the following error caused by inertia, and the notch filter was designed in the PMAC motion controller. The main function of the notch filter prevented the system from resonating and canceled the resonance in this way. Figure 12 illustrates the PID controller parameter adjustment principle provided by the PMAC motion control card.

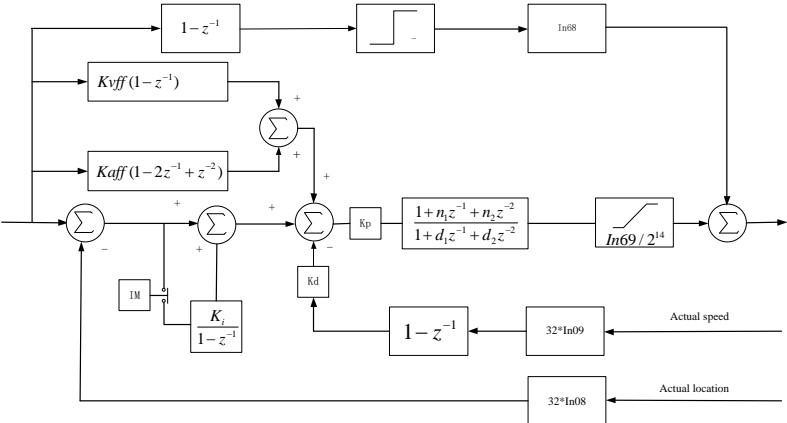

**Figure 12.** Schematic diagram of the proportion–integral–derivative (PID) controller parameter adjustment in the PMAC motion control card.

Optimizing parameters such as $K_p$, $K_d$, $K_i$, $K_{vff}$, *IM*, and $K_{aff}$ was the main task of the PMAC motion controller. In Figure 11, $K_p$ is the proportional gain; $K_d$ is the differential gain; $K_{vff}$ is the speed feed forward gain; $K_i$ is the integral gain; *IM* is the integral mode coefficient; and $K_{aff}$ is the acceleration feed forward gain. The variables P, I, and D mean the proportion part, integral part, and derivative part of proportion–integral–derivative (PID).

## 6. Simulations and Experiments

*6.1. Statistical Analysis of the Humanoid Robotic Arm Massage System*

The ANSYS software is a general finite element analysis software. To verify whether the design of the robot arm strength and stiffness meet the design requirements, the humanoid massage robot arm was built via CATIA 3D model with the data interface built into the ANSYS [29,30]. A stress diagram and displacement nephogram can be obtained, which evaluates the strength and stiffness of the parts and components of the whole arm structure optimization, and achieves miniaturization, weight reduction, and anthropomorphism of the massage robot arm.

A humanoid massage robot arm has a series structure, where the arm shell, the rotating shaft, and the bearing may be deformed. The rigidity of the arm indicates the end-effector, which can overcome the deformation force under the external force. The strength and stiffness of the humanoid massage robot arm are the main factors which affect the dynamic characteristics and position accuracy under loaded conditions. Therefore, it is a key step in designing a humanoid massage robot arm. The size of the external load directly determines the size of the terminal deformation, which affects the positioning accuracy of the terminal. In a certain range of terminal variation, the static stiffness can be controlled by the terminal output force.

During the movement of the humanoid massage robot arm, there will be a state of maximum force and deformation, that is, the arms of the robot are in a straight line. At this time, the gravity arm reaches the maximum value, so each arm of the humanoid massage robot arm is subject to the maximum state of force. In this paper, the state of maximum force and deformation were selected and investigated for static analysis. Both the upper and lower arms were fixed at one end, and the other end applied a reaction force and bending moment when the constraints were added in the design of the humanoid massage robot arm. Thereby, we could obtain the deformation and stress of each component in the most dangerous and deformed state. Owing to the load type and the structural characteristics and analysis requirements of the built entity, the size of the mesh division should be less than or equal to the minimum wall thickness of the entity. Finally, we adopted a free meshing method for the body of the humanoid massage robot arm. The following were the specific mesh divisions: the boom bracket was divided into 8223 units and 37,357 nodes; the shoulder pitch bracket was divided into 5127 units and 21,767 nodes; the number of units obtained by the elbow pitch bracket was 4772, and the number of nodes was 18,155. The elbow rotation bracket was divided into 4462 units and 15,076 nodes.

The finite element analysis of the key components of the shoulder joint and the elbow joint was completed by adding a load, and the torque was applied by establishing a node at the center of the component, which obtained a highly accurate simulation result. The stress and deformation cloud diagrams of the shoulder and elbow joint structures are shown in Figures 13 and 14.

The stress and deformation diagrams show that the structural design was unreasonable and facilitated the determination of corresponding improvements. The key components on the arm of the humanoid massage robot were analyzed in the actual analysis through the ANSYS software. The design of each part ensured rigidity to minimize weight. Therefore, the designed arm was also optimized by using dynamics and finite element analysis. The small deformation area of the arm was improved, resulting in a small, lightweight, and anthropomorphic arm.

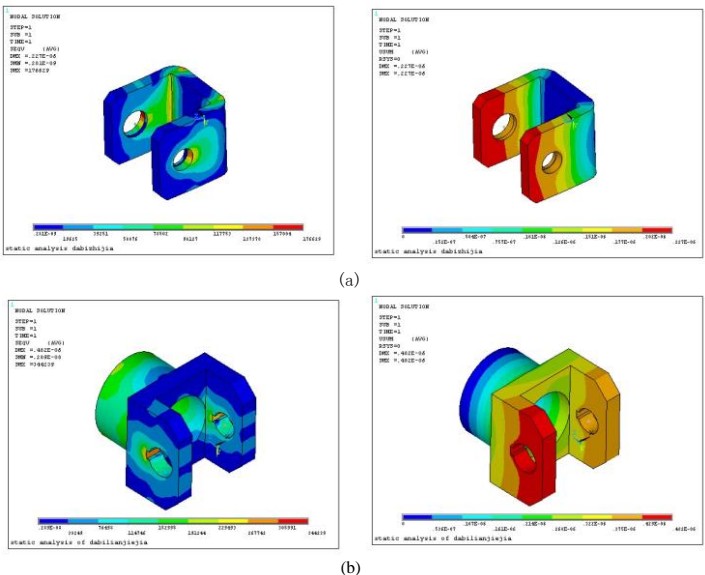

**Figure 13.** Shoulder joint stress and deformation cloud map. (**a**) Schematic diagrams of stress and deformation of the shoulder bracket. (**b**) Schematic diagrams of stress and deformation of the shoulder pitching bracket structure.

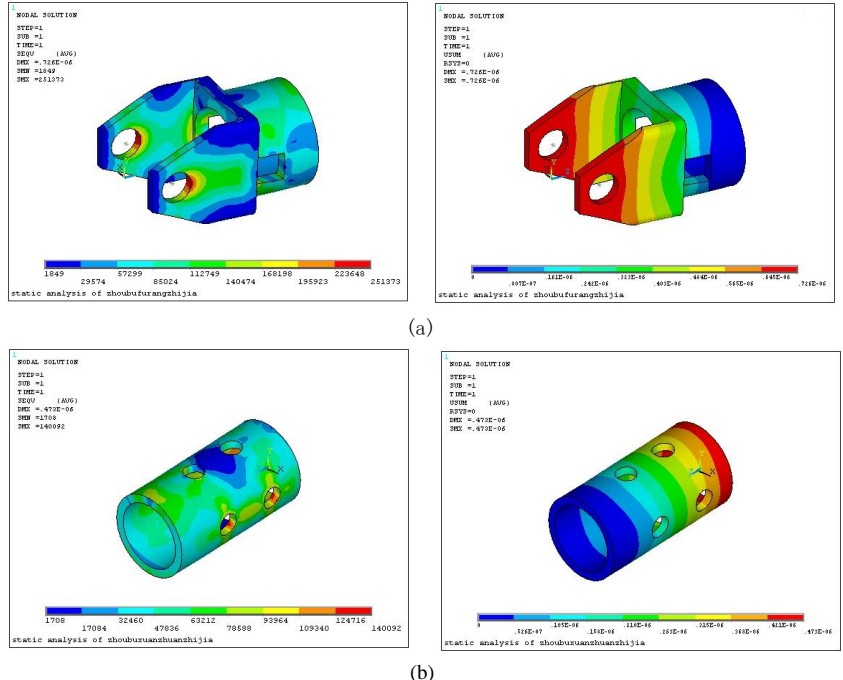

**Figure 14.** Elbow joint stress and deformation cloud map. (**a**) Stress and deformation cloud diagram of the elbow pitch bracket structure. (**b**) Structural stress and deformation cloud diagram of the elbow rotation bracket.

## *6.2. Kinematic Simulation in ADAMS Environment*

To view the actual operation of the humanoid massage robot arm, the kinematics analysis was carried out and investigated in the design process. The model built in CATIA software was imported into ADAMS software. The motion curves were observed to satisfy the design requirements. Kinematic analysis was a critical basis of the mechanical design, performance analysis, and control of the humanoid massage robot. The analysis also provided a reference for the improvement of the arm design and adjusted the transmission mechanism of the human machine. The 3D modeling and

interference analysis of the arm structure of the humanoid massage robot were realized through CATIA software. Subsequently, using ADAMS software, the mechanical dynamics model was simulated by the SimDesigner interface to achieve the seamless connection between the patient and massage arm robot. The simulation system can be built using various joints of inertia, rotation speed, torque, and force conditions, as well as other parameters. The motor and reducer models were appropriately selected for the massage arm robot model. The driving function of each joint can be described as follows: the pitch joint function of upper arm was −90d × sin (45 × time), the terminal time was 25 s, and the step was 200. The rotation joint function of elbow was 180d × sin (time), the terminal time was 25 s, and the step was 200. The pitch function of the wrist was 90d × sin (time), the terminal time was 25 s, and the step was 200.

Figure 15 illustrates the curves of the different joints. As can be observed, the motion of each arm was smooth, realistic, and in accordance with the experimental requirements, verifying the feasibility of the constructed humanoid massage robot arm system theoretically. In particular, the design of the physical prototype was based on the requirements of the maximum torque of each joint. It thereby guarantees the reliability of the design and provides the reference for the design, optimization, and motion control of the humanoid robotic arm massage system.

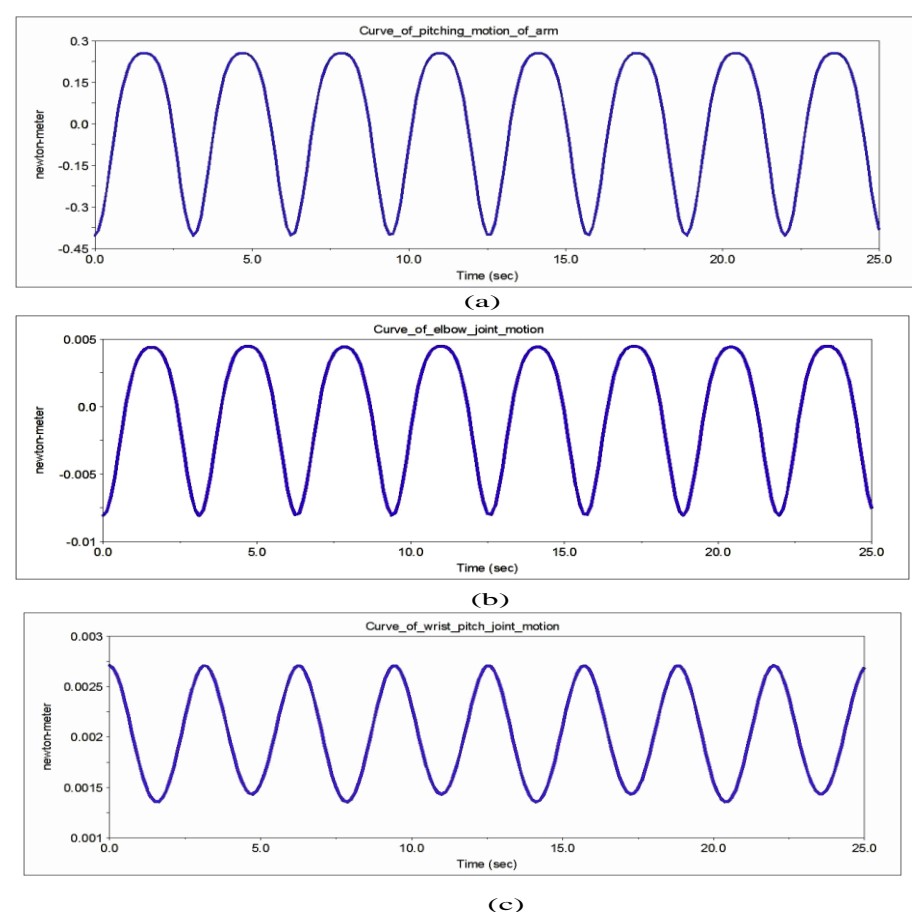

**Figure 15.** Plots of the different joint motions. (**a**) Curve of pitching motion of the arm. (**b**) Curve of the elbow joint motion. (**c**) Curve of the wrist pitch joint motion.

In this paper, the periodic motion of the single join was analyzed for the humanoid massage robot arm, and then the curves of the pitching joint, elbow joint, and wrist joint were obtained as shown in Figure 16.

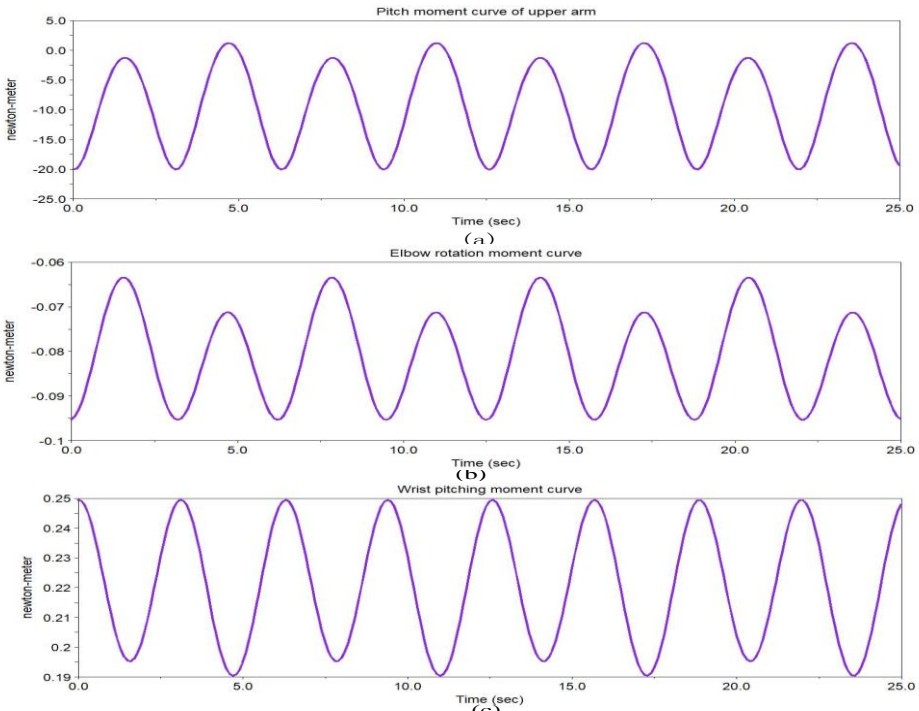

**Figure 16.** Periodic curves of single joints of the humanoid massage robot arm. (**a**) The curve of the arm pitching joint. (**b**) The curve of elbow movement. (**c**) The curve of wrist movement.

*6.3. Experimental Verification*

6.3.1. Experiments on Repeated Positioning Accuracy

The repeatability of positioning accuracy was an important parameter for the humanoid massage arm robot. The parameters of the humanoid massage arm robot were measured to verify the characteristics of joint motions under a full load. Owing to human trajectories, the Chinese medicinal massage arm robot could utilize 10 kg as a full load at the end of the humanoid massage arm robot. It was required to return to the zero point of the *x*-axis and proceed from this point to a specified position. The actual movement distance was measured using a Vernier caliper. The positional accuracy of the humanoid Chinese massage robot arm was recorded and calculated during the process of rehabilitation training through repetitive experiments.

More specifically, PMAC PEWIN32 software was utilized to return the *x*-axis back to zero. When it reached zero, the position was marked. The forward command of the *x*-axis was entered, and the command arm of the humanoid massage robot arm moved forward by 10 mm. A Vernier caliper with the precision of 0.02 mm was utilized to measure the distance of the relative zero-mark position of the human body. The *x*-axis returned back to the zero instructions again. At the same time, the forward command of the *x*-axis was adopted once more, and the command arm of the Chinese medicinal massage robot moved forward by 100 mm. The distance of the relative zero-mark position of the humanoid massage robot was measured upon the completion of movement. This experiment was repeated several times, and the data are tabulated in Table 3. Remarkably, as for the *x*-axis, the maximum error of positional accuracy was less than 0.1 mm, which satisfied the requirement of precision for the improved movement of the Chinese medicinal humanoid massage robot arm.

**Table 3.** Repetitive positioning accuracy of experimental data.

| Number of Experiments | Actual Running Distance/mm | Error/mm |
|:---:|:---:|:---:|
| 1 | 200.032 | 0.032 |
| 2 | −200.070 | 0.070 |
| 3 | 200.026 | 0.026 |
| 4 | −199.986 | −0.014 |
| 5 | 200.096 | 0.096 |
| 6 | −199.994 | −0.006 |
| 7 | 200.068 | 0.068 |
| 8 | −200.028 | 0.028 |
| 9 | 200.030 | 0.030 |
| 10 | −199.960 | −0.040 |

### 6.3.2. Precision Experiments on Massage Strength

The control strength of massage force, which is a constant of the arm, was controlled by the force control of the arm-kneading method. The control precision of massage force was tested during the process of rehabilitation training. A palm rub massage force control experiment was performed (rehabilitation removed) to create a massage robot as a platform. In accordance with the requirements of the palm kneading movement, palm massage operations in the *x*- and *y*-axes were carried out to realize massage trajectory, the *x*-axis was utilized to exert massage force, and the massage contact points were used for the end of the parallel massage head. The force signal was amplified and put into the corresponding AD channel of the PMAC motion control card. The card then regulated the massage force according to the skin model-based massage force control algorithm program.

Owing to the designed requirements, the control accuracy of force should be less than 5% for the improved massage robot, i.e., the robot's massage force should not exceed 50 N. Hence, the resolution ratio of electronic measuring equipment should be less than 0.005 kg and the measuring range should be 20 kg, as shown in Figure 17.

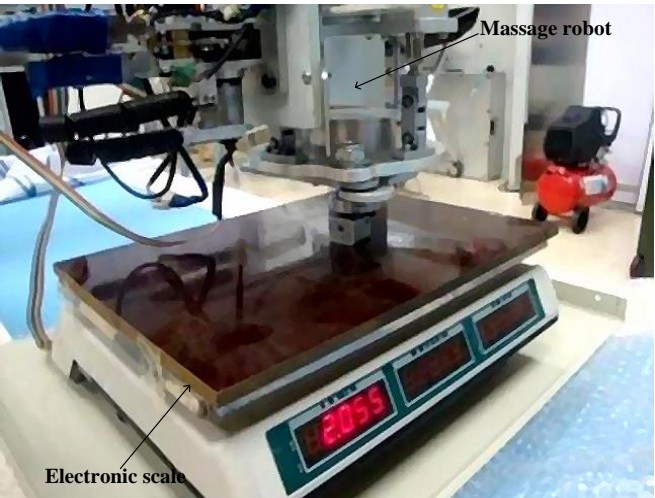

**Figure 17.** Massage force precision experimental platform.

The experimental process was referred to an object to be massaged and was utilized to display the actual massage force. Prior to the test, the parallel massage head mechanism was perpendicular to the electronic scale and displayed no contact with the electronic scale. The expected force of the palm rubbing method was manually entered to stabilize after the robot movement. The actual strength was recorded, and the desired force was altered. Furthermore, the measurement was performed again. Note that the expected force ranged from 1 kg to 4 kg during the total of seven experiments.

The experimental data are shown in Table 4. It was not difficult to calculate a force error of less than 5%, satisfying the required strength for an improved force of the massage robot.

**Table 4.** Experimental data on the control of massage force.

| Experiment Number | Expectation Force | Measured Force | Force Deviation | Force Accuracy |
|:---:|:---:|:---:|:---:|:---:|
| 1 | 1.00 | 1.05 | 0.05 | 5% |
| 2 | 1.50 | 1.43 | −0.07 | 4.6% |
| 3 | 2.00 | 2.07 | 0.07 | 3.5% |
| 4 | 2.50 | 2.57 | 0.07 | 2.8% |
| 5 | 3.00 | 2.92 | −0.08 | 2.7% |
| 6 | 3.50 | 3.40 | −0.10 | 2.9% |
| 7 | 4.00 | 4.10 | 0.10 | 2.5% |

*6.4. Simulation Results of Massage Robot Arm for PID Controller*

To verify the feasibility and efficiency of the PID controller, the massage robot system can be seen in Figure 18. The angular velocity and angular acceleration response curve of the massage robot arm boom pitch joint are shown in Figure 19a. The angular acceleration curve of the massage robot arm is in blue, and the angular velocity curve of the massage robot arm is in red.

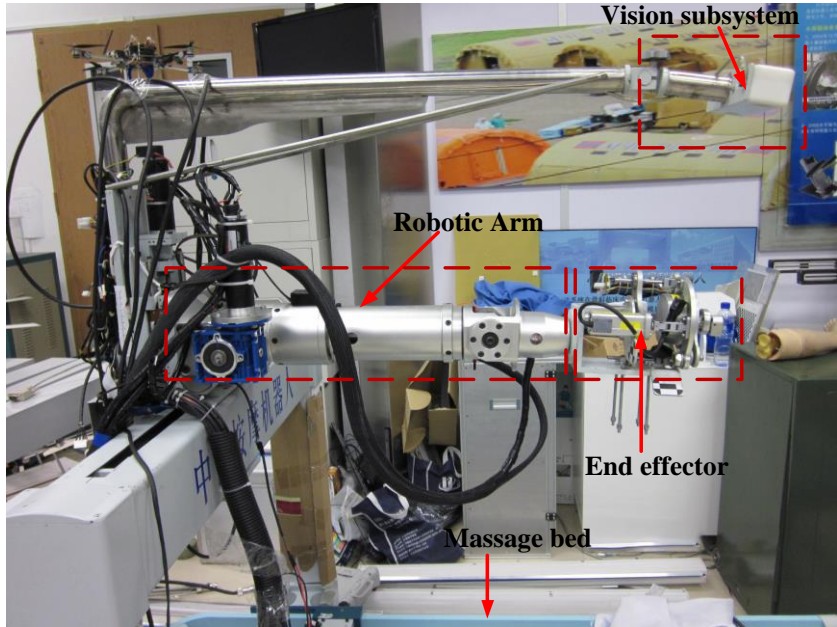

**Figure 18.** Massage Robot System.

It can be seen from Figure 16b,c that the massage robot arm ran smoothly. In the designed trajectory, it could operate according to the expected trajectory and the temporary overshoot amount was small in the step signal. In addition, the response time was short and the PID in the position controller was used. The effect was obvious when the parameters were adjusted to the appropriate parameter values.

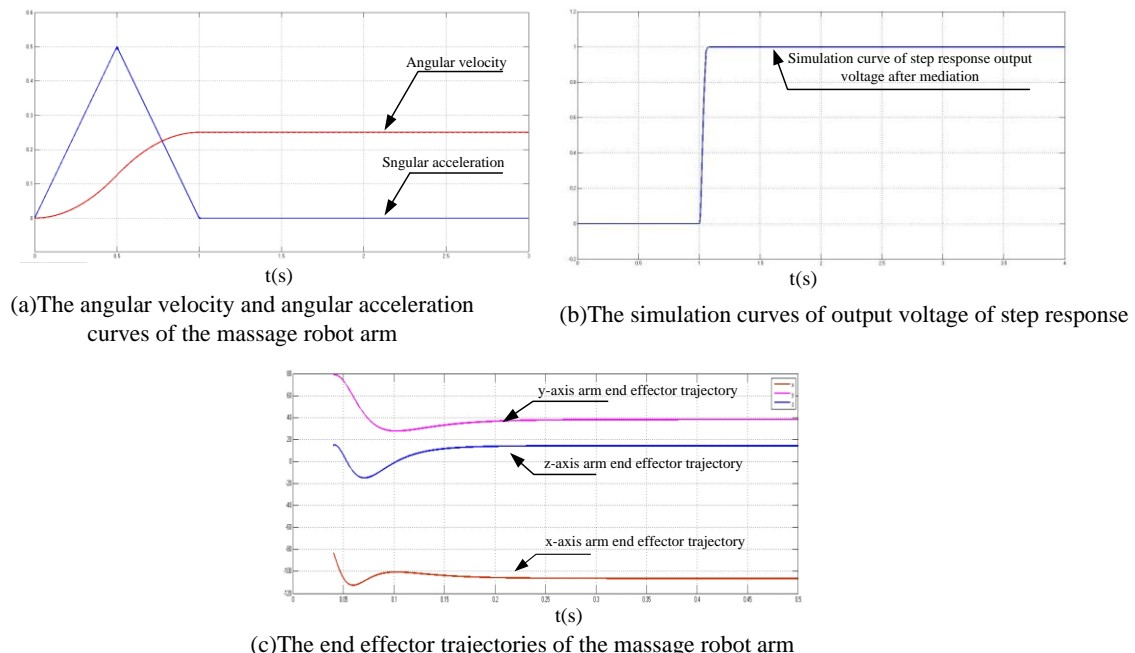

(a)The angular velocity and angular acceleration curves of the massage robot arm

(b)The simulation curves of output voltage of step response

(c)The end effector trajectories of the massage robot arm

**Figure 19.** Different simulation results of the massage robot arm.

## 7. Discussion

In this paper, the effectiveness of the working mechanism of the Chinese medical massage arm robot was numerically and experimentally investigated. Specifically, according to the functional and control requirements, the arm of a Chinese massage robot was proposed and developed for the process of rehabilitation training. The working space was analyzed via the SimMechanics toolbox in MATLAB software, and the finite element and dynamic simulation analyses were combined to improve the massage robot. An adaptive controller was designed and analyzed for the Chinese massage robot, and the trajectories of the end-effector were described in relation to rehabilitation training activities. The repetitive position, massage strength, and massage effects of precision experiments were also obtained for the process of rehabilitation training. Numerical simulations and experimental results revealed that the designed arm possessed a compact structure, high precision, a light weight, and good rigidity, which satisfied the requirements and improved the carrying capacity of the arm. Remarkably, the humanoid massage robot arm's performance was stable and reliable.

In future work, the structure's optimization will be conducted to make the massage robot arm more compact and anthropomorphic. Furthermore, the control algorithm of the humanoid massage robot arm will be extensively investigated to enhance the control precision and interaction ability of the robot system. Eventually, advancements in humanoid massage robots will in turn provide the impetus for the development of rehabilitation medical equipment in the context of Chinese medicinal massage.

## 8. Conclusions

In conclusion, a compact, lightweight, and rigid human-like massage robot arm was designed, investigated, and analyzed for natural rehabilitation of patients. The kinematics and workspace analysis were carried out for a massage robot arm. The combination of finite element analysis and dynamic simulation analysis was utilized to improve it. A PID controller and the numerical simulation method were used to complete the design of the arm control system. Experiments such as repeated positioning accuracy, massage force precision, and massage effect were considered and analyzed for the natural rehabilitation of patients. The experimental results verified that the designed arm satisfied the design requirements and provided a design method for the development of Chinese massage robots.

**Author Contributions:** In this work, Z.P., B.Z., and J.Y. conceived and designed the experiments; Z.S. performed the experiments; Z.P. and Z.S. analyzed the data; L.G. contributed the analysis tools; and Z.P. wrote the paper.

**Funding:** This work was supported in part by the National Natural Science Foundation of China under Grant 61751304, Grant 51875047, and Grant 61873304, and in part by the China Postdoctoral Science Foundation funded under Grant 2019T120240 and 2018M641784, and in part by the Jilin Province Development and Reform Commission under Grant 2018C037-1.

**Conflicts of Interest:** The authors declare no conflict of interest.

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
