# Peer review of "Design and Analysis of a Chinese Medicine Based Humanoid Robotic Arm Massage System"

_applsci, doi:10.3390/app9204294_

Round 1

Reviewer 1 Report

This paper presents the design and analysis of a robotic arm for human body massage purposes.

The proposed work is based on existing developments and systems as stated by the authors (lines 54-70) in the literature (refs 2-24). Therefore, it is not clear which is the novelty or contribution of this work compared to other similar works presented in introduction.

I would suggest to provide a separate section/paragraph about the related work compared with more details against the presented system, in order to explain better the advantages of the proposed work.

Some other suggestions follow:

Pay attention to how citing references according to the journal format, e.g. in line 58 Minyong et al. [ref?]

The claimed contributions of this work (lines 100-107) should be clearly and comprehensive. The second contribution of this robotic arm for degenerative diseases and chronic pains is somewhat not proved in practice yet. About the third claimed contribution, actually only the shoulder is examined for strength and stiffness. At least another key component, as it is the end-effector (details for which actually is not provided), is not examined.

The authors present the kinematics analysis of the proposed 4DOF humanoid robotic arm, although this analysis does not present anything new to the proposed approach, beyond the already known information about forward and inverse kinematics analysis provided in the literature.

They do not provide any dynamics analysis of the proposed robotic arm mechanism, neither of the moving object (human body under control, I suppose). I do not insist that there is definitely a necessity.

There is not any information or description details about the end-effector. What end-effector is utilized?

I would avoid to use the term “humanoid” to describe the proposed robotic arm mechanism, since does not possess at least a humanoid-alike dexterous robotic hand, as the end-effector, however I could accept it in the description.

In line 254 the authors state that the mechanical system of the humanoid massage robotic arm is modeled by Sim Mechanics toolbox in MATLAB software. In Figure 8 it is not clear whether the 3D solid model of the shoulder is created using the Sim Mechanics toolbox in MATLAB or the ANSYS software mentioned later.

Because in line 334 the authors state that the solid model of humanoid Chinese massage robot arm is created in the 3D modeling software which is loaded into ANSYS software. So, the purpose of utilizing the ANSYS software is not clearly explained at this point.

Avoid to use abbreviations in abstract without explanation, e.g. in line 285 PMAC abbreviation should be defined.

In line 307 the authors state that the PWIN32 software is adopted to set and debug the PID parameters of the joint, and in line 310 that the computer adjusts the corresponding I variable through the executable program PEWIN32. Please clarify these abbreviations and the ‘I’ variable.

In line 335 reference is required for ANSYS software.

In section 6.2 Kinematic simulation in ADAMS environment, in line 361 the authors state that the 3D modeling and interference analysis of the arm structure of humanoid massage robot are realized through CATIA software.

(i) Reference is required for CATIA software too.

(ii) It is not clear the use of CATIA as another 3D modelling software, since MATLAB SIM Mechanics and ANSYS software have already been used for 3D modelling and analysis of the robotic arm.

(iii) It is not clear, what kind of interference analysis of the arm structure of humanoid massage robot is realized with CATIA software.

(iv) In line 363 authors state that using ADAMS software, the mechanical dynamics model is simulated by SimDesigner. Once again, it is not clarified the necessity of another 3D modelling and simulation tool, and what are the differences with the other 3D modelling and simulation tools already used.

(v) This section is supposed to be about kinematics analysis, not dynamics. Anyway, the authors do not provide any dynamic analysis of the robotic arm mechanism, nor of the object (body) under control in this paper.

Author Response

Revision Information (#Ref. No.: applsci-608900)

“Design and Analysis of a Chinese Medicine Based Humanoid Robotic Arm Massage System”

Zaixiang Pang

Dear Editors and Reviewers,

On behalf of all the contributing authors, I would like to express our sincere appreciation for your letter and reviewers’ constructive comments concerning our article entitled “Design and Analysis of a Chinese Medicine Based Humanoid Robotic Arm Massage System” (Manuscript No.: applsci-608900).

According to these comments, we have made extensive modifications to our manuscript and supplemented extra simulations to make our results convincing. Point-by-point responses to the comments of associate editor and reviewers are listed below with the corresponding revisions highlighted in blue colour in the revised manuscript.

Response to Reviewer 1 Comments

Point 1: The proposed work is based on existing developments and systems as stated by the authors (lines 54-70) in the literature (refs 2-24). Therefore, it is not clear which is the novelty or contribution of this work compared to other similar works presented in introduction.

I would suggest to provide a separate section/paragraph about the related work compared with more details against the presented system, in order to explain better the advantages of the proposed work.

Response 1: The authors sincerely thank the reviewer for raising such a comment. We have made correction according to the reviewer’s comments.

In this paper, the humanoid massage robot arm is mainly used in the field of traditional Chinese medicine massage to realize the massage of the lumbar spine and back. According to the task function and control requirements, the humanoid massage robot arms are designed with joint as the module based on the characteristics of human skeleton and the massage theory of traditional Chinese medicine, which compared with the previous humanoid massage robot arms. The joints of humanoid massage robot arm are connected together to realize the basic functions of human-like arm by taking advantage of the working space and good flexibility of the serial robot, thus, the massage terminal and the massage hand are delivered to the precise massage position.

(1) The mechanical structure of humanoid massage robot arm is developed, investigated and analysed through the modular design concept, which is composed of shoulder joint, elbow rotatory joint, wrist swing and wrist rotatory joint. First, the motor of shoulder joint is installed perpendicular to the axis of the arm, which increases the motion space of the arm and provides convenience for replacing the motor. Second, the U-like structure is adopted in the shoulder bracket to improve the bearing capacity of the arm. Third, the motor (which installed in the arm) is coincident with the centre line of the arm to improve the transmission precision and reduce the diameter of each joint. Therefore, the characteristics of humanoid massage robot arm are compact structure and anthropomorphism, which assembly reduces the accumulated error of arm. The bevel gear is utilized to swinging joint, which improves the stability of the humanoid massage robot arm. Servo motor is used to drive each joint of the humanoid massage robot arm, and the reducer and brake are installed to realize overload protection and emergency braking.

(2) The strength and stiffness of humanoid massage robot arm are analysed to verify that the mechanical mechanism satisfies the design requirements of mechanics. The kinematics simulation of the humanoid massage robot arm is illustrated and carried out through the establishment of virtual prototype model, which provides a reference for the improvement of the humanoid massage robot arm.

(3) A three-loop servo system is established, investigated and verified for the humanoid massage robot arm. The smooth operation and trajectory positioning of the humanoid massage robot arm are realized by adjusting the PID controller of the humanoid massage robot arm to adjust the parameters and combining with the PMAC motion controller.

Point 2: Pay attention to how citing references according to the journal format, e.g. in line 58 Minyong et al. [ref?].

Response 2: The authors sincerely thank the reviewer for the time, effort and the frank recognition given to the manuscript. We have made correction according to the reviewer’s comments.

Point 3: The claimed contributions of this work (lines 100-107) should be clearly and comprehensive.  The second contribution of this robotic arm for degenerative diseases and chronic pains is somewhat not proved in practice yet.  About the third claimed contribution, actually only the shoulder is examined for strength and stiffness. At least another key component, as it is the end-effector (details for which actually is not provided), is not examined.

Response 3: Thank you very much for raising this suggestion. The details can be seen as follows:

The second contribution: The strength and stiffness of humanoid massage robot arm are analysed to verify that the mechanical mechanism satisfies the design requirements of mechanics. The kinematics simulation of the humanoid massage robot arm is illustrated and carried out through the establishment of virtual prototype model, which provides a reference for the improvement of the humanoid massage robot arm.

The third contribution: A three-loop servo system is established, investigated and verified for the humanoid massage robot arm. The smooth operation and trajectory positioning of the humanoid massage robot arm are realized by adjusting the PID controller of the humanoid massage robot arm to adjust the parameters and combining with the PMAC motion controller.

The stress and deformation cloud diagram of the elbow pitch bracket structure and the structural stress and deformation cloud diagram of the elbow rotation bracket are shown in the figure 13.

(a) Schematic diagrams of stress and deformation of shoulder bracket. (b) Schematic diagrams of stress and deformation of shoulder pitching bracket structure.

Figure 13. Shoulder joint stress and deformation cloud map

Point 4: They do not provide any dynamics analysis of the proposed robotic arm mechanism, neither of the moving object (human body under control, I suppose). I do not insist that there is definitely a necessity.

Response 4: The authors sincerely thank the reviewer for such a helpful suggestion. We have made explanation according to the reviewer’s comments. The details can be seen as follows:

To view the actual operation of the humanoid massage robot arm, the kinematics analysis is carried out and investigated in the design process. The motion curves are observed to satisfy the design requirements. The analysis of the dynamics model is mainly analyzed for the force of the humanoid massage robot arm during the movement. The dynamics of the humanoid massage robot arm and the control technique under the dynamics framework are not discussed in this paper.  

Point 5: There is not any information or description details about the end-effector. What end-effector is utilized?

Response 5: The authors sincerely thank the reviewer for the time, effort and suggestions spent in reviewing our manuscript.

The end-effector of the humanoid massage robot arm mainly consists of a three-finger massage dexterous hand for the manipulation of kneading and a miniaturized parallel robot massage terminal for the manipulation of pressing, kneading, vibration and rolling. The massage marker points need to wear vest, and prone on the massage bed through the binocular visual positioning sensors before implementation of massage. The movement of humanoid massage robot arm drives the dexterous hand massage and parallels robot terminal to the location, and combines with the arm front terminal and massages the hand of human body.

Fig 9 The end-effector of the humanoid massage robot arm

Point 6: In line 254 the authors state that the mechanical system of the humanoid massage robotic arm is modelled by Sim Mechanics toolbox in MATLAB software. In Figure 8 it is not clear whether the 3D solid model of the shoulder is created using the Sim Mechanics toolbox in MATLAB or the ANSYS software mentioned later.

Response 6: The authors sincerely thank the reviewer for such a helpful suggestion. We have made

correction according to the reviewer’s comments. The details can be seen as follows:

To reasonably allocate dimensions, ensure the coordination of dimensions and the smooth progress, it is necessary to check that whether the parts touch or block each other, which may result in the failure of normal installation. In addition, whether the positioning and assembly relationship between parts is reasonable, and whether the overall structure is beautiful should be considered for humanoid massage robot arm. The 3D solid modelling of humanoid massage robot arm is carried out and investigated based on CATIA 3D modelling software, which realized parametric modelling and virtual assembly of humanoid massage robot arm from top to bottom. It can establish an accurate analysis model for finite element analysis, which improves the speed and accuracy of design requirements.

Point 7: Because in line 334 the authors state that the solid model of humanoid Chinese massage robot arm is created in the 3D modelling software which is loaded into ANSYS software. So, the purpose of utilizing the ANSYS software is not clearly explained at this point.

Response 7: The authors sincerely thank the reviewer for his/her frank recognition given to the manuscript, in addition to the time and effort spent in reviewing our manuscript.

ANSYS software is a general finite element analysis software. To verify whether the design of the robot arm strength and stiffness meet the design requirements, the humanoid massage robot arm is built via CATIA 3D model with the data interface into the ANSYS. It can be obtained the stress diagram and displacement nephogram, which evaluates the strength and stiffness of the parts and components to the whole arm structure optimization, and achieves to miniaturization, lightweight, anthropomorphic umanoid massage robot arm.

Point 8: Avoid to use abbreviations in abstract without explanation, e.g. in line 285 PMAC abbreviation should be defined.

Response 8: The authors sincerely thank the reviewer for the time, effort and suggestions spent in reviewing our manuscript. The details can be seen as follows:

Programmable Multi-Axis Controller referred to as "PMAC" is a programmable multi-axis motion controller.

Point 9: In line 307 the authors state that the PWIN32 software is adopted to set and debug the PID parameters of the joint, and in line 310 that the computer adjusts the corresponding I variable through the executable program PEWIN32. Please clarify these abbreviations and the ‘I’ variable.

Response 9: The authors sincerely thank the reviewer for the time, effort and suggestions spent in reviewing our manuscript. The details can be seen as follows:

The variable “I” means the Integral part of Proportion-Integral-Derivative (PID);

Point 10: In line 335 reference is required for ANSYS software.

Response 10: The authors sincerely thank the reviewer for the time, effort and suggestions spent in reviewing our manuscript. We have made correction according to the reviewer’s comments.

Point 11: In section 6.2 Kinematic simulation in ADAMS environment, in line 361 the authors state that the 3D modelling and interference analysis of the arm structure of humanoid massage robot are realized through CATIA software.

Response 11: The authors sincerely thank the reviewer for such a helpful suggestion. We have made correction according to the reviewer’s comments. The details can be seen as follows:

(1) To figure out the working space at the end of the massage mechanical arm with the angle of each joint as input function, the corresponding system model is established based on MATLAB SIM Mechanics toolbox, and was conducted through the motion simulation.

(2) To check the actual operation of the humanoid massage robot arm, the model built in CATIA software is imported into Adams software. Therefore, the actual motion curves of each joints of the humanoid massage robot arm are obtained to observe whether it satisfies the designing requirements.

Finally, we would like to say thanks again sincerely to the editor and anonymous reviewers for their time and effort spent in handling our manuscript, as well as providing us many constructive comments for improving very much the presentation and quality of this manuscript. According to reviewers’ suggestions, we tried our best to improve the manuscript and made some changes in the manuscript. These changes will not influence the content and framework of the paper. We appreciate for Editors’ warm work earnestly and hope that the correction will meet with approval. Once again, thank you very much for your comments and suggestions.

Reviewer 2 Report

This paper deals with a design and analysis of a Chinese Medicine based humanoid robotic arm massage system. This paper is well organized through whole paper and well express the previous work in introduction. However, I would like to point out following as: 1. In introduction, this paper is ambiguous the object and subject. I hope authors should add more detailed and clear object of this paper, why this paper is necessary in the side of reader. 2. I wander why author should not consider dynamic phenomena? 3. In Fig 13, to verify exact motion, I think authors should add result of circular motion. 4. I hope authors add following reference. (1) Byoung-Ho Kim, Analysis on Load Torque Effect for Assistive Robotic Arms International Journal of Fuzzy Logic and Intelligent Systems, Vol. 18, No. 4, 2018, pp. 276-283. http://doi.org/10.5391/IJFIS.2018.18.4.276 (2) Akinori Wakabayashi, Satona Motomura, and Shohei Kato, Associative Motion Generation for Humanoid Robot Reflecting Human Body Movement, International Journal of Fuzzy Logic and Intelligent Systems, vol. 12, no. 2, June 2012, pp. 121-130 http://dx.doi.org/10.5391/IJFIS.2012.12.2.121

Author Response

Revision Information (#Ref. No.: applsci-608900)

“Design and Analysis of a Chinese Medicine Based Humanoid Robotic Arm Massage System”

Zaixiang Pang

Dear Editors and Reviewers,

On behalf of all the contributing authors, I would like to express our sincere appreciation for your letter and reviewers’ constructive comments concerning our article entitled “Design and Analysis of a Chinese Medicine Based Humanoid Robotic Arm Massage System” (Manuscript No.: applsci-608900).

According to these comments, we have made extensive modifications to our manuscript and supplemented extra simulations to make our results convincing. Point-by-point responses to the comments of associate editor and reviewers are listed below with the corresponding revisions highlighted in blue colour in the revised manuscript.

Response to Reviewer 2 Comments

Point 1: 1. In introduction, this paper is ambiguous the object and subject. I hope authors should add more detailed and clear object of this paper, why this paper is necessary in the side of reader.

Response 1: The authors sincerely thank the reviewer for raising such a comment. We have made correction according to the reviewer’s comments.

In this paper, the humanoid massage robot arm is mainly used in the field of traditional Chinese medicine massage to realize the massage of the lumbar spine and back. According to the task function and control requirements, the humanoid massage robot arms are designed with joint as the module based on the characteristics of human skeleton and the massage theory of traditional Chinese medicine, which compared with the previous humanoid massage robot arms. The joints of humanoid massage robot arm are connected together to realize the basic functions of human-like arm by taking advantage of the working space and good flexibility of the serial robot, thus, the massage terminal and the massage hand are delivered to the precise massage position.

(1) The mechanical structure of humanoid massage robot arm is developed, investigated and analysed through the modular design concept, which is composed of shoulder joint, elbow rotatory joint, wrist swing and wrist rotatory joint. First, the motor of shoulder joint is installed perpendicular to the axis of the arm, which increases the motion space of the arm and provides convenience for replacing the motor. Second, the U-like structure is adopted in the shoulder bracket to improve the bearing capacity of the arm. Third, the motor (which installed in the arm) is coincident with the centre line of the arm to improve the transmission precision and reduce the diameter of each joint. Therefore, the characteristics of humanoid massage robot arm are compact structure and anthropomorphism, which assembly reduces the accumulated error of arm. The bevel gear is utilized to swinging joint, which improves the stability of the humanoid massage robot arm. Servo motor is used to drive each joint of the humanoid massage robot arm, and the reducer and brake are installed to realize overload protection and emergency braking.

(2) The strength and stiffness of humanoid massage robot arm are analysed to verify that the mechanical mechanism satisfies the design requirements of mechanics. The kinematics simulation of the humanoid massage robot arm is illustrated and carried out through the establishment of virtual prototype model, which provides a reference for the improvement of the humanoid massage robot arm.

(3) A three-loop servo system is established, investigated and verified for the humanoid massage robot arm. The smooth operation and trajectory positioning of the humanoid massage robot arm are realized by adjusting the PID controller of the humanoid massage robot arm to adjust the parameters and combining with the PMAC motion controller.

Point 2: I wander why author should not consider dynamic phenomena?

Response 2: The authors sincerely thank the reviewer for such a helpful suggestion. We have made explanation according to the reviewer’s comments. The details can be seen as follows:

To view the actual operation of the humanoid massage robot arm, the kinematics analysis is carried out and investigated in the design process. The motion curves are observed to satisfy the design requirements. The analysis of the dynamics model is mainly analyzed for the force of the humanoid massage robot arm during the movement. The dynamics of the humanoid massage robot arm and the control technique under the dynamics framework are not discussed in this paper.

Point 3: In Fig 13, to verify exact motion, I think authors should add result of circular motion.

Response 3:  The authors sincerely thank the reviewer for such a helpful suggestion. We have made explanation according to the reviewer’s comments.

In this paper, the periodic motion of the single join is analyzed for the humanoid massage robot arm, and then the moment curves of the pitching joint, elbow joint and wrist joint are obtained as shown in Figure 16.

(a)The curve of arm pitching joint (b) The curve of elbow moment (c) The curve of wrist moment

Figure 16. Periodic curves of single joint of humanoid massage robot arm

Point 4: I hope authors add following reference.

(1) Byoung-Ho Kim, Analysis on Load Torque Effect for Assistive Robotic Arms International Journal of Fuzzy Logic and Intelligent Systems, Vol. 18, No. 4, 2018, pp. 276-283. http://doi.org/10.5391/IJFIS.2018.18.4.276

(2) Akinori Wakabayashi, Satona Motomura, and Shohei Kato, Associative Motion Generation for Humanoid Robot Reflecting Human Body Movement, International Journal of Fuzzy Logic and Intelligent Systems, vol. 12, no. 2, June 2012, pp. 121-130 http://dx.doi.org/10.5391/IJFIS.2012.12.2.121

Response 4: The authors sincerely thank the reviewer for the time, effort and suggestions spent in reviewing our manuscript. We have made correction according to the reviewer’s comments.

Finally, we would like to say thanks again sincerely to the editor and anonymous reviewers for their time and effort spent in handling our manuscript, as well as providing us many constructive comments for improving very much the presentation and quality of this manuscript. According to reviewers’ suggestions, we tried our best to improve the manuscript and made some changes in the manuscript. These changes will not influence the content and framework of the paper. We appreciate for Editors’ warm work earnestly and hope that the correction will meet with approval. Once again, thank you very much for your comments and suggestions.

Round 2

Reviewer 1 Report

The authors have responded sufficiently to most of the comments.

Just a brief notice. Although in my first comment (Point 1) they have responded sufficiently with additional paragraphs explaining more in detail the advantages of their proposed work, however, I would expect to see this additional information presented in a comparative way against any related work (for which additional references could also be provided, or could be used some of the references already stated from 2 to 24, in other words, try to avoid to group many references and better try to explain which is the contribution of each one).

Anyway, the above are not compulsory. If the authors could improve them a little bit further, that would be nice. Otherwise, I could say that I am satisfied with the current version, thus I can recommend it for publication.

Author Response

Revision Information (#Ref. No.: applsci-608900)

“Design and Analysis of a Chinese Medicine Based Humanoid Robotic Arm Massage System”

Zaixiang Pang

Dear Editors and Reviewers,

On behalf of all the contributing authors, I would like to express our sincere appreciation for your letter and reviewers’ constructive comments concerning our article entitled “Design and Analysis of a Chinese Medicine Based Humanoid Robotic Arm Massage System” (Manuscript No.: applsci-608900).

According to these comments, we have made extensive modifications to our manuscript and supplemented extra simulations to make our results convincing. Point-by-point responses to the comments of associate editor and reviewers are listed below with the corresponding revisions highlighted in blue colour in the revised manuscript.

Response to Reviewer 1 Comments

Point 1: Just a brief notice. Although in my first comment (Point 1) they have responded sufficiently with additional paragraphs explaining more in detail the advantages of their proposed work, however, I would expect to see this additional information presented in a comparative way against any related work (for which additional references could also be provided, or could be used some of the references already stated from 2 to 24, in other words, try to avoid to group many references and better try to explain which is the contribution of each one).

Anyway, the above are not compulsory. If the authors could improve them a little bit further, that would be nice. Otherwise, I could say that I am satisfied with the current version, thus I can recommend it for publication.

Response 1: The authors sincerely thank the reviewer for raising such a comment. We have made correction according to the reviewer’s comments.

Many companies and institutions carried out relevant research about massage robotic systems. The massage service equipment has been developed which expanded the scope of care and treatment of massage [2]. The force control was investigated and analyzed for the path planning of induction-based massage robot [3]. Moreover, a 3-degree of freedoms (DOFs) massaging robot was designed and investigated for the patients [4]. Chinese traditional massage mainly emphasizes the stimulation of manipulation and changing power on a series of meridians which improves the ability of the immune system and achieves the purpose of disease prevention and treatment. There are lots of works which. extensively studied a finger and 13 joints of the human finger multi-finger massage robot [5,6,7]. The existing massage devices and massage robots which are relatively single, and can be realized only one or two different massage techniques generally study and design certain massage techniques for patients. Subsequently, a novel PUMA562 robot platform was developed and designed for the people [8]. Recently, some classical massage robots have been developed, such as WAO-1 [9,10], humanoid massage robot [11], four-finger human hand robot [12], and robotic hand [13]. To relieve lumbago and leg pain for middle/old aged degenerative, the Chinese-style massage robot was designed and investigated on the basis of Chinese medicine massage theory [14,15,16]. The massage force can be actively controlled through the force sensors, however, the personalized massage cannot be implemented for patients. Moreover, Hu et al. developed and analyzed a Chinese massage robot, which featured an arm with four DOFs [17]. This robot had the end of its arm in series with three degrees of freedom, and parallel wrist 3 served to the human hand massage. Additionally, two fingers are ineffective, and the other fingers can make actions such as clicking, finger kneading, pinching, and rolling. Therefore, the traditional five kinds of actions which are the Chinese medicine techniques could be performed [18]. In addition, more and more researchers study humanoid robot arms from the perspective of service robots [19,20,21,22,23,24]. Therefore, the Chinese traditional massage devices and massage robots can be adjusted passively by the masseur or the massage object, which is difficult to ensure the accuracy and reliability of the massage position.

In recent years much attention has been paid to the massage equipment and massage robotic systems, which can be applied to many fields such as health care and rehabilitation therapy. The technical and functional characteristics of massage robot arm are as follows: 1) relatively high degree of humanity and 2) ability for remarkably complex movements and accurate grasp of massage force. In particular, the humanoid massage robot arm is rarely focused on configuration for realizing various massage techniques and precise recognition of designated massage position. Furthermore, during the performance estimation of humanoid Chinese medicine massage robotic arm system, many factors including the weight of the arm and massage force size, direction, location, acupuncture point, and frequency should be taken into account. Therefore, a robotic arm with rigidity, high transmission accuracy, and anthropomorphic degree is highly desirable for the patients.

To address the above-mentioned shortage on medical personnel and improve the quality of life of the elderly and the disabled, a humanoid robot massage arm is designed on the basis of human arm massage theory. In this paper, the humanoid massage robot arm is mainly used in the field of traditional Chinese medicine massage to realize the massage of the lumbar spine and back. According to the task function and control requirements, the humanoid massage robot arms are designed with joint as the module based on the characteristics of human skeleton and the massage theory of traditional Chinese medicine, which compared with the previous humanoid massage robot arms. The joints of humanoid massage robot arm are connected together to realize the basic functions of human-like arm by taking advantage of the working space and good flexibility of the serial robot. The binocular vision positioning technology is utilized to determine the precise massage position. Furthermore, the three-finger massage dexterous hand and parallel robot massage are used to reproduce the techniques of experts. Moreover, the movement of the robot arm drives the dexterous hand and the parallel robot massage terminal to the designated position. In addition, the humanoid massage robot massages the patient through the massage terminal at the front of the arm and the massage hand. Therefore, it can achieve the massage techniques (which can be considered as press, knead, knead, roll, vibration) of human lumbar spine and the back of the palm which stimulates meridian points and improves the ability of the immune system. Therefore, the natural posture of the arm of humanoid Chinese medicine massage robot is an important indicator of massage robotic system [25]. Four degrees of freedom of humanoid Chinese medicine massage robotic arm (through serial structure in series with the elbow, forearm, and wrist joints) can easily achieve the action of the arm.

Finally, we would like to say thanks again sincerely to the editor and anonymous reviewers for their time and effort spent in handling our manuscript, as well as providing us many constructive comments for improving very much the presentation and quality of this manuscript. According to reviewers’ suggestions, we tried our best to improve the manuscript and made some changes in the manuscript. These changes will not influence the content and framework of the paper. We appreciate for Editors’ warm work earnestly and hope that the correction will meet with approval. Once again, thank you very much for your comments and suggestions.

Reviewer 2 Report

I think this paper well revised according to reviewer's point out. Thus I would like to decide as an "accept"

Author Response

Revision Information (#Ref. No.: applsci-608900)

“Design and Analysis of a Chinese Medicine Based Humanoid Robotic Arm Massage System”

Zaixiang Pang

Dear Editors and Reviewers,

On behalf of all the contributing authors, I would like to express our sincere appreciation for your letter and reviewers’ constructive comments concerning our article entitled “Design and Analysis of a Chinese Medicine Based Humanoid Robotic Arm Massage System” (Manuscript No.: applsci-608900).

Response to Reviewer 2 Comments

Point 1: I think this paper well revised according to reviewer's point out. Thus I would like to decide as an "accept"

Response 1: We would like to say thanks again sincerely to the editor and anonymous reviewers for their time and effort spent in handling our manuscript, as well as providing us many constructive comments for improving very much the presentation and quality of this manuscript. We appreciate for Editors’ warm work earnestly and hope that the correction will meet with approval. Once again, thank you very much for your comments and suggestions.
